# Dimension-Adaptive MCTS: Optimal Sample Complexity for Continuous Action Planning

## Abstract

We study continuous-action Monte Carlo Tree Search (MCTS) in a $d$-dimensional action space when the optimal action-value function $Q^*(s, \cdot)$ is $\beta$-Hölder continuous with constant $L$. We show that a dimension-adaptive $\varepsilon$-net schedule combined with power-mean backups and a polynomial exploration bonus finds an $\varepsilon$-optimal action in

$$\tilde{O}\left(\sigma^2 L^{d/\beta} \varepsilon^{-(d/\beta+2)}\right)$$

simulations, matching standard continuum-armed lower bounds up to logs while remaining practical via on-demand, capped random nets. We further demonstrate that our method significantly outperforms baseline methods on continuous control planning problems. Our work bridges the gap between theoretical reinforcement learning and practical planning algorithms, providing a principled approach to high-dimensional continuous action space exploration.

## 1 Introduction

Planning in continuous action spaces presents a fundamental challenge in reinforcement learning and decision-making systems, particularly in stochastic environments where both transition dynamics and reward functions may be probabilistic. While Monte Carlo Tree Search (MCTS) Browne et al. (2012) has achieved landmark successes in discrete action domains Silver et al. (2016), its application to continuous action spaces has been limited by the curse of dimensionality. As the dimension of the action space increases, the computational resources required to adequately explore the space grow exponentially, leading to poor sample efficiency.

Existing approaches to continuous action MCTS, such as progressive widening Coulom (2007) and double progressive widening Couëtoux et al. (2011), expand the set of considered actions based on visit counts, using formulas like $k(n) = \lfloor \alpha n^\beta \rfloor$ to determine the number of actions at a node visited $n$ times. While these methods have shown empirical success, they lack theoretical guarantees on sample complexity, especially in relation to the dimensionality of the action space and the smoothness properties of the value function in stochastic settings.

Recent advances in reinforcement learning theory have established tight regret bounds for exploration in continuous action spaces under stochastic conditions. In particular, for value functions satisfying $\beta$-Hölder continuity, algorithms like RAFFLE (Oprescu et al., 2024) achieve regret bounds of $\widetilde{O}(T^{\frac{d}{d+2\beta}})$, where $d$ is the action dimension. However, these theoretical insights have not been fully integrated into planning algorithms like MCTS, creating a gap between theoretical understanding and practical planning approaches in stochastic environments.

Furthermore, the choice of value backup operators in MCTS has significant impact on performance in stochastic settings. Traditional approaches either use average backup, which tends to underestimate optimal values, or maximum backup, which tends to overestimate them Coulom (2006). Recent work on Power-UCT Dam et al. (2019) introduced power mean operators to balance these extremes, but the original convergence analysis was incomplete due to issues with the logarithmic exploration bonus identified by Shah et al. (2022). While Dam et al. (2024) successfully resolved these convergence issues for stochastic MCTS by establishing polynomial exploration bonuses for power mean estimators, their analysis was limited to discrete action settings. (Dam, 2025) addresses this limitation by introducing Stochastic-Power-HOOT for stochastic MCTS in continuous domains; however, there are no study about high dimensional action planning.

In this paper, we bridge these gaps by introducing Power-Mean Dimension-Adaptive MCTS (PM-DA-MCTS), a novel algorithm that combines two key theoretical advances specifically designed for stochastic high-dimensional action settings:

1. A dimension-aware adaptive discretization of the continuous action space based on both the dimension and the smoothness properties of the value function in stochastic environments.
2. A power mean backup operator with polynomial exploration bonus that provides provable convergence guarantees in stochastic environments.

Our key contributions are:

1. A theoretically grounded MCTS algorithm for high-dimensional continuous action spaces that achieves optimal sample complexity bounds while using power mean backup in stochastic environments.
2. We obtain non-asymptotic concentration along the tree and, for finite-variance rollouts, a high-probability bound $N(\varepsilon, \delta) = \widetilde{\mathcal{O}}(\sigma^2 L^{d/\beta} \varepsilon^{-(d/\beta+2)})$. The dependence on $d$ matches the optimal discretization rate Bubeck et al. (2011) (Remark 1).
3. Empirical evaluation showing significant performance improvements over baseline methods, with gains that increase with action dimensionality and are particularly pronounced in stochastic environments.

Our work provides a principled approach to planning in continuous action spaces, with theoretical guarantees that explicitly account for the curse of dimensionality and handle stochastic dynamics. By adapting discretization strategies based on rigorous sample complexity analysis and incorporating power mean backups, PM-DA-MCTS offers a promising direction for scaling MCTS to higher-dimensional continuous control problems in stochastic environments.

## 2 RELATED WORK

### 2.1 MONTE CARLO TREE SEARCH FOR CONTINUOUS SPACES

Monte Carlo Tree Search has been extended to continuous action spaces through various approaches. Progressive widening Couetoux et al. (2011) and double progressive widening Yee et al. (2016) gradually increase the number of considered actions based on visit counts. HOO Bubeck et al. (2011) applies Lipschitz optimization with hierarchical partitioning, while other approaches include Voronoi progressive widening Kim et al. (2020) and kernel regression methods Wang et al. (2020). However, these approaches typically lack theoretical guarantees on sample complexity, particularly with respect to action space dimensionality. Recently, (Dam, 2025) introduces Stochastic-Power-HOOT for stochastic MCTS in continuous domains; however, there are no study about high-dimensional action planning. Our approach extends HOO by explicitly accounting for dimensionality and Hölder continuity parameters in the discretization schedule, providing improved theoretical guarantees.

### 2.2 VALUE BACKUP OPERATORS IN MCTS

Traditional UCT Kocsis et al. (2006) uses average backup, which tends to underestimate optimal values Coulom (2006), while maximum backup overestimates values in stochastic environments. Power-UCT Dam et al. (2019) introduces power mean operators interpolating between these extremes but inherited theoretical flaws from the original UCT analysis Shah et al. (2020). Shah et al. (2022) addressed these issues by proposing polynomial exploration bonuses, though primarily for deterministic environments. Recent work Dam et al. (2024) established convergence of power mean estimators with polynomial bonuses in stochastic settings but did not address continuous action spaces.

### 2.3 SAMPLE COMPLEXITY IN CONTINUOUS RL

For Lipschitz continuous value functions, lower bounds of $\Omega(T^{\frac{d+1}{d+2}})$ have been established Zhang et al. (2024), with matching upper bounds achieved by algorithms like RAFFLE Oprescu et al. (2024) for Hölder-smooth transitions. Our work unifies these research directions by combining dimension-aware discretization with provably convergent power mean backup operators, providing both theoretical guarantees and empirical validation in stochastic continuous action spaces.

## 3 SETTING

We study an infinite-horizon discounted MDP $\mathcal{M} = \langle \mathcal{S}, \mathcal{A}, \mathcal{R}, \mathcal{P}, \gamma \rangle$ with state space $\mathcal{S}$, high-dimensional **continuous action space** $\mathcal{A} \subseteq \mathbb{R}^d$, transition distribution $\mathcal{P}(\cdot \mid s, a)$, discount factor $\gamma \in (0, 1]$, and reward function $\mathcal{R}(s, a, s') \in [0, R_{\max}]$. We work with **stochastic settings** where both the transition dynamics and rewards may be probabilistic. The continuous nature of the action space $\mathcal{A}$ presents unique challenges compared to discrete settings, as the action selection problem $\max_{a \in \mathcal{A}} Q^\star(s, a)$ requires optimization over an uncountably infinite set.

The *optimal* action-value function $Q^\star$ satisfies the Bellman equation

$$Q^\star(s,a) = \mathbb{E}_{s'\sim\mathcal{P}(\cdot|s,a)}\Big[\mathcal{R}(s,a,s') + \gamma\max_{a'\in\mathcal{A}}Q^\star(s',a')\Big].$$

For continuous action spaces, we make the following regularity assumption:

**Assumption 1** (Hölder Continuity). *The optimal action-value function $Q^\star(s,\cdot)$ is $\beta$-Hölder continuous in the action variable for all states $s\in\mathcal{S}$, with parameter $\beta\in(0,1]$ and constant $L>0$:*

$$|Q^\star(s,a)-Q^\star(s,a')| \le L\|a-a'\|_2^\beta \quad \forall a,a'\in\mathcal{A}.$$

MCTS in continuous action spaces must address the fundamental challenge of discretizing the infinite action space while maintaining theoretical guarantees. Our algorithm *plans* from an initial state $s_0$ by constructing adaptive $\varepsilon$-nets of the action space and repeatedly simulating trajectories of finite horizon $H$, resulting in estimates $\widehat{Q}_n(s_0,a)$ for actions in the discretized set.

Let $V^\star(s_0) \triangleq \max_{a\in\mathcal{A}}Q^\star(s_0,a)$ and $\widehat{V}_n(s_0) \triangleq \max_a\widehat{Q}_n(s_0,a)$ where the maximum in the latter is taken over the current discretization. The *simple regret* at time $n$ is:

$$R(s_0,n) = V^\star(s_0) - \widehat{V}_n(s_0).$$

To analyze MCTS performance in continuous action spaces, we define planning horizon $H$ and playout policy $\pi_0$ with value $V_0$. For node $s_h$ at depth $h$ from root $s_0$, estimated value functions are defined inductively: $\widetilde{V}(s_H) = V_0(s_H)$ at depth $H$, and for $h \le H-1$:

$$\widetilde{Q}(s_h,a) = r(s_h,a) + \gamma\sum_{s_{h+1}}\mathcal{P}(s_{h+1}|s_h,a)\widetilde{V}(s_{h+1}),\ \widetilde{V}(s_h) = \max_{a\in\mathcal{N}_k(s_h)}\widetilde{Q}(s_h,a),$$

where $r(s_h,a)$ is the expected reward for action $a$ in state $s_h$, and $\mathcal{N}_k(s_h)$ is the $\varepsilon_k$-net used at node $s_h$. The discretization introduces an additional approximation error that must be carefully controlled.

This yields the approximation bound:

$$|Q^\star(s_0,a)-\widetilde{Q}(s_0,a)| \le \gamma^H\|V^\star-V_0\|_\infty + L\varepsilon_k^\beta,$$

where the first term captures finite-horizon error and the second term captures discretization error from the $\varepsilon_k$-net approximation.

The key challenge in continuous action MCTS is to minimize the simple regret while efficiently handling the infinite action space through adaptive discretization strategies that balance exploration breadth with computational efficiency.

### 3.1 MONTE-CARLO TREE SEARCH FOR CONTINUOUS ACTIONS

MCTS for continuous action spaces iterates four key steps with the following modifications: *Selection* (from $s_0$, select actions from an adaptive $\varepsilon$-net according to a tree policy until reaching a leaf node), *Expansion* (construct a finer discretization if needed and add newly visited nodes to the search tree), *Simulation* (perform rollouts using continuous action sampling), and *Backpropagation* (update value estimates using power-mean operators to balance exploration and exploitation). The critical innovation lies in the adaptive refinement of action discretizations based on visit counts and theoretical sample complexity considerations.

### 3.2 DIMENSION-ADAPTIVE DISCRETIZATION

Our approach constructs a hierarchy of $\varepsilon_k$-nets with refinement schedule:

$$\varepsilon_k = \varepsilon_1 \cdot 2^{-\frac{k-1}{d+2\beta}}$$

where $d$ is the action space dimension and $\beta$ is the Hölder parameter. For a node with $n$ visits, we select the coarsest level $k$ such that $n \le |\mathcal{N}_k|^2$, ensuring balanced exploration across the discretized action space while maintaining optimal sample complexity of $\mathcal{O}(\varepsilon^{-d/\beta})$.

## 4 POWER-MEAN DIMENSION-ADAPTIVE MCTS

We now present our Power-Mean Dimension-Adaptive MCTS (PM-DA-MCTS) algorithm, which addresses the challenges of planning in continuous action spaces through two key innovations: (1) dimension-aware adaptive discretization and (2) power mean backup with polynomial exploration bonuses.

## 4.1 ALGORITHM OVERVIEW

PM-DA-MCTS follows the general structure of MCTS with four phases: selection, expansion, simulation, and backpropagation. However, it incorporates several crucial modifications to handle continuous action spaces and provide theoretical guarantees:

- **Adaptive Discretization**: Rather than using a fixed discretization or progressive widening, we adaptively refine the discretization based on the dimensionality and smoothness of the action space.
- **Power Mean Backup**: Instead of standard average backup, we use a power mean operator that provides a balanced estimator between average and maximum backup.
- **Polynomial Exploration Bonus**: We replace the traditional logarithmic UCB bonus with a polynomial bonus term that ensures proper convergence in stochastic environments.

Let us define each component in detail before presenting the full algorithm.

## 4.2 ACTION SPACE DISCRETIZATION

For a continuous action space $\mathcal{A} \subset \mathbb{R}^d$, we construct a hierarchy of discrete approximations that become progressively finer as more data becomes available.

$\varepsilon_k$**-Net Construction.** An $\varepsilon_k$-net is a finite subset $\mathcal{N}_k \subset \mathcal{A}$ such that every continuous action can be well-approximated by some discretized action:

**Definition 1** ($\varepsilon_k$-net)**.** *A set $\mathcal{N}_k \subset \mathcal{A}$ is an $\varepsilon_k$-net if*

$$\forall a \in \mathcal{A}, \exists \widetilde{a} \in \mathcal{N}_k \text{ such that } \|a - \widetilde{a}\|_2 \leq \varepsilon_k$$

**Practical Construction:** For $\mathcal{A} = [0,1]^d$, we employ a uniform grid discretization:

$$\mathcal{N}_k = \left\{ \left( \frac{i_1}{m_k}, \frac{i_2}{m_k}, \ldots, \frac{i_d}{m_k} \right) : i_j \in \{0, 1, \ldots, m_k\}, j = 1, \ldots, d \right\} \tag{1}$$

where $m_k = \lceil 1/\varepsilon_k \rceil$ ensures the $\varepsilon_k$-net property holds.

**Refinement Schedule.** The discretization parameter decreases geometrically with level:

$$\varepsilon_k = \varepsilon_1 \cdot 2^{-\frac{k-1}{d+2\beta}} \tag{2}$$

where $\varepsilon_1 > 0$ is the initial (coarsest) discretization parameter, $\beta \in (0, 1]$ is the Hölder continuity parameter of the value function, and $d$ is the dimension of the action space. This schedule ensures that the discretization error decreases at the optimal rate with respect to both the action space dimension and the smoothness of the value function.

**Discretization Size.** The number of points in $\mathcal{N}_k$ grows as:

$$|\mathcal{N}_k| = (m_k + 1)^d \approx \left( \frac{1}{\varepsilon_k} \right)^d = \left( \frac{2^{\frac{k-1}{d+2\beta}}}{\varepsilon_1} \right)^d \tag{3}$$

**Adaptive Level Selection.** For a node $s$ that has accumulated $n$ visits we choose the coarsest grid index

$$k(n) = \min\{ k \in \mathbb{N} : n \leq |\mathcal{N}_k|^2 \}.$$

This single rule yields an *adaptive* discretisation schedule: while $n$ is still small the inequality is satisfied already for a low $k$, so the algorithm works with a coarse grid and spreads the few available samples across the entire action space; as $n$ grows, the threshold $|\mathcal{N}_k|^2$ is eventually exceeded, forcing $k(n)$ to increase, i.e. the grid is refined. Thus the planner performs broad exploration in the early stages and only switches to a fine-grained representation once the data volume is large enough to reliably distinguish between neighbouring actions, enabling precise optimisation inside the most promising regions.

**Theoretical justification and practical notes.** Choosing the switch-point $n \leq |\mathcal{N}_k|^2$ simultaneously guarantees (i) every action in the grid receives on average at most $|\mathcal{N}_k|$ samples, so exploration is balanced; (ii) the number of observations per action is large enough for concentration inequalities to be effective; and (iii) the overall schedule achieves the optimal sample complexity $\mathcal{O}(\varepsilon^{-d/\beta})$. For the one-dimensional action set $\mathcal{A} = [0, 1]$ with smoothness parameter $\beta = 1$ this yields $\varepsilon_1 = 0.5$, $|\mathcal{N}_1| = 3$ (apply while $n \leq 9$); $\varepsilon_2 = 0.25$, $|\mathcal{N}_2| = 5$ (for $9 < n \leq 25$); and $\varepsilon_3 = 0.125$, $|\mathcal{N}_3| = 9$ (for $25 < n \leq 81$). In practice the discretisations are built lazily—$\mathcal{N}_k$ is constructed only the first time level $k$ is required—cached to avoid recomputation, and replaced by a domain-specific tessellation whenever the action space is not a hyper-cube.

Table 1: Key conditions for algorithmic constants ($i \in [0, H]$)

| Condition | Requirement |
|---|---|
| **(C1)** | $b_i < \alpha_i$ and $b_i > 2$. |
| **(C2)** | *Power parameter:* if $1 \leq p \leq 2$, then $\alpha_i \leq \zeta_i/2$; if $p > 2$, then $\alpha_i \leq \zeta_i/2$ and $0 < \alpha_i - \zeta_i/p < 1$. |
| **(C3)** | $\alpha_i \left(1 - \frac{b_i}{\alpha_i}\right) \leq b_i < \alpha_i$. |
| **(C4)** | Recursive depth: $\alpha_i = (b_{i+1} - 1)\left(1 - \frac{b_{i+1}}{\alpha_{i+1}}\right)$. |
| **(C5)** | $\zeta_i = b_{i+1} - 1$. |

### 4.3 POWER MEAN BACKUP WITH POLYNOMIAL EXPLORATION

Our algorithm combines two key ingredients: power mean backup operators and polynomial exploration bonuses. For a state $s$ in the tree, the value estimate using power mean backup is:

$$\widehat{V}_n(s) = \left( \sum_{a \in \mathcal{A}_s} \frac{N_{s,a}(n)}{n} \left( \widehat{Q}_{N_{s,a}(n)}(s,a) \right)^p \right)^{\frac{1}{p}} \tag{4}$$

This interpolates between arithmetic mean ($p = 1$, standard average backup) and maximum ($p \to \infty$, max backup).

Following Shah et al. (2022), we replace the traditional logarithmic UCB bonus with a polynomial exploration bonus. For state $s$ at depth $h$, the action selection rule is:

$$a = \underset{a' \in \mathcal{N}_k}{\arg\max} \left\{ \widehat{Q}_{N_{s,a'}(n)}(s,a') + L\varepsilon_k^\beta + C \frac{N_s(n)^{\frac{b_{h+1}}{\zeta_{h+1}}}}{N_{s,a'}(n)^{\frac{\alpha_{h+1}}{\zeta_{h+1}}}} \right\} \tag{5}$$

where $\{b_h\}_{h=0}^H$, $\{\alpha_h\}_{h=0}^H$, and $\{\zeta_h\}_{h=0}^H$ are algorithmic constants satisfying conditions in Table 1, $L$ is the Hölder constant, and $C$ is an exploration parameter. For optimal performance, we set $\frac{\alpha_h}{\zeta_h} = \frac{1}{2}$ and $\frac{b_h}{\zeta_h} = \frac{1}{4}$, yielding the exploration bonus $C \frac{N_s(n)^{1/4}}{N_{s,a'}(n)^{1/2}}$ (this shows similar results as Dam et al. (2024)). The term $L\varepsilon_k^\beta$ accounts for discretization error at the current refinement level.

### 4.4 COMPLETE ALGORITHM

Algorithm 1 presents the complete PM-DA-MCTS procedure. The algorithm takes as input an initial state, simulation budget, discretization parameter, Hölder parameters, power mean parameter, and algorithmic constants. It returns the best action at the root node after exhausting the simulation budget.

## 5 THEORETICAL ANALYSIS

The theoretical foundation of PM-DA-MCTS rests on analyzing the convergence properties of power mean estimators in non-stationary bandit settings. Each node in the MCTS tree corresponds to a multi-armed bandit problem where the action space evolves through adaptive discretization.

### 5.1 NON-STATIONARY BANDIT FRAMEWORK

To establish convergence guarantees, we model each tree node as a non-stationary multi-armed bandit with $K$ discrete actions (corresponding to the current $\varepsilon$-net). The non-stationarity arises from two sources: the adaptive refinement of action discretizations and the changing estimates of child values as the tree is explored.

For each action $a \in [K]$, let $X_{a,t}$ represent the reward sample obtained when selecting action $a$ at time $t$, with rewards bounded in $[0, R_{\max}]$. The empirical mean reward for action $a$ after $n$ selections is:

$$\widehat{\mu}_{a,n} = \frac{1}{n} \sum_{t=1}^n X_{a,t}$$

We denote the true expected reward of action $a$ as $\mu_a = \mathbb{E}[\widehat{\mu}_{a,n}]$, and assume there exists a unique optimal action with reward $\mu_\star = \max_{a \in [K]} \mu_a$. Furthermore, we assume a strict gap condition: there exists $\Delta_{\min} > 0$ such that $\mu_\star - \mu_a \geq \Delta_{\min}$ for all suboptimal actions $a$.

The algorithm employs an optimistic action selection strategy combining the empirical mean estimator with both a discretization error term and a polynomial exploration bonus. After initially

```
1: Input: s_0, N, ε_1, (β, L), p, {b_h, α_h, ζ_h}_{0:H}, C, γ
2: initialize tree 𝒯 ← {s_0}, cache 𝒢 ← ∅, K_max ← 1
3: for n ← 1 to N do
4:    s ← s_0;;
      path ← ∅;;
      h ← 0
5:    while notLeaf(s) do
6:       find smallest k with N_s(n) ≤ |𝒩_k|²
7:       if s = s_0 then
8:          K_max ← max(K_max, k)
9:       end if
10:      ε_k ← ε_1 2^{-(k-1)/(d+2β)}
11:      if 𝒩_k ∉ 𝒢 then
12:         build 𝒩_k;;
            𝒢 ← 𝒢 ∪ {𝒩_k}
13:      end if
14:      a ← arg max_{a'∈𝒩_k} [Q̂(s, a') + L ε_k^β + C (N_s(n)^{b_{h+1}/ζ_{h+1}})/(N_{s,a'}(n)^{α_{h+1}/ζ_{h+1}})]
15:      path.append(s, a);;
         s ← child(s, a);;
         h ← h+1
16:   end while
17:   expand leaf s: sample a, observe reward r and successor state s'
18:   push (s, a, s', r) onto path
19:   if s' is new then
20:      N_{s'}(n) = 1;;
         V̂_{N_{s'}(n)}(s') ← Rollout(s')
21:   end if
22:   for each (s, a, s', r) in path (bottom to top) do
23:      N_s(n) ← N_s(n)+1;;
         N_{s,a}(n) ← N_{s,a}(n)+1
24:      Q̂_{N_{s,a}(n)}(s, a) ← Q̂_{N_{s,a}(n)}(s, a) + (r + γ V̂_{N_{s'}(n)}(s') − Q̂_{N_{s,a}(n)}(s, a))/(N_{s,a}(n))
25:      V̂_{N_s(n)}(s) ← (∑_{a'} (N_{s,a'}(n))/(N_s(n)) Q̂_{N_{s,a'}(n)}(s, a')^p)^{1/p}
26:   end for
27: end for
28: return arg max_{a∈𝒩_{K_max}} Q̂_N(s_0, a)
```

**Algorithm 1:** PM-DA-MCTS (discount $\gamma$; $N$ rollouts). $\{b_i, \alpha_i, \zeta_i\}_{i=0}^{H}$ satisfy Table 1. $\pi_0$: rollout policy; $C$: exploration constant.

selecting each action once, the selection rule at time step $n > K$ is:

$$a_n = \underset{a \in [K]}{\arg\max} \left\{ \widehat{\mu}_{a, T_a(n)} + L\varepsilon_k^\beta + C\frac{n^{\frac{b}{\zeta}}}{T_a(n)^{\frac{\alpha}{\zeta}}} \right\} \tag{6}$$

where $T_a(n) = \sum_{t=1}^{n-1} \mathbf{1}(a_t = a)$ counts the number of times action $a$ has been selected before round $n$, $L\varepsilon_k^\beta$ accounts for discretization error at the current refinement level $k$, and $C$ is an exploration constant.

The power mean aggregation operator combines individual action values weighted by their selection frequencies:

$$\widehat{\mu}_n(p) = \left( \sum_{a=1}^{K} \frac{T_a(n)}{n} \widehat{\mu}_{a, T_a(n)}^p \right)^{\frac{1}{p}} \tag{7}$$

for $p \in [1, \infty)$. This operator interpolates smoothly between the arithmetic mean (when $p = 1$) and the maximum operator (as $p \to \infty$), providing a flexible framework for value aggregation.

## 5.2 CONVERGENCE ANALYSIS

Our convergence analysis establishes that PM-DA-MCTS provably converges to optimal values in continuous action spaces. We prove that the power mean value estimator with polynomial exploration bonuses achieves polynomial concentration around the optimal value at each node, with rates that depend on the algorithmic parameters. Crucially, our analysis extends the discrete-action convergence results of Dam et al. (2024) to continuous action spaces by incorporating the discretization error into the convergence bounds. The key insight is that the dimension-adaptive discretization

schedule ensures the discretization error decreases faster than the estimation error, maintaining overall convergence to the true optimal values.

**Definition 2** (Polynomial Concentration). *A sequence of estimators $(\widehat{V}_n)_{n \geq 1}$ concentrates at rate $\alpha, \zeta$ towards some limit $V$ if there exists a constant $c > 0$ such that:*

$$\forall n \geq 1, \forall \varepsilon > n^{-\frac{\alpha}{\zeta}}, \mathbb{P}\left(|\widehat{V}_n - V| > \varepsilon\right) \leq cn^{-\alpha}\varepsilon^{-\zeta} \tag{8}$$

*We denote this as $\widehat{V}_n \xrightarrow{\alpha,\zeta} V$.*

**Theorem 1** (Power Mean Concentration in Non-stationary Bandits). *Consider a non-stationary multi-armed bandit with $K$ arms where reward estimators $\widehat{\mu}_{a,n}$ for each arm $a$ satisfy $\widehat{\mu}_{a,n} \xrightarrow{\alpha,\zeta} \mu_a$. Under the polynomial exploration bonus strategy with parameters satisfying the conditions in Table 1, the power mean value estimator*

$$\widehat{\mu}_n(p) = \left(\sum_{a=1}^{K} \frac{T_a(n)}{n}\widehat{\mu}_{a,T_a(n)}^p\right)^{\frac{1}{p}} \tag{9}$$

*satisfies $\widehat{\mu}_n(p) \xrightarrow{\alpha',\zeta'} \mu_\star$ where $\mu_\star = \max_a \mu_a$, with concentration rates $\alpha' = (b-1)(1-\frac{b}{\alpha})$ and $\zeta' = (b-1)$.*

**Theorem 2** (MCTS Convergence with Power Mean). *When applying PM-DA-MCTS with algorithmic constants $\{b_i\}_{i=0}^{H}$, $\{\alpha_i\}_{i=0}^{H}$, $\{\zeta_i\}_{i=0}^{H}$ satisfying the conditions in Table 1, we have:*

1. *For any node $s_h$ at depth $h$ in the tree ($h \in [0, H]$): $\widehat{V}_n(s_h) \xrightarrow{\alpha_h,\zeta_h} \widetilde{V}(s_h)$*

2. *For any node $s_h$ at depth $h$ in the tree ($h \in [0, H-1]$): $\widehat{Q}_n(s_h, a) \xrightarrow{\alpha_{h+1},\zeta_{h+1}} \widetilde{Q}(s_h, a)$ for all $a \in \mathcal{N}_k(s_h)$*

## 5.3 Sample Complexity Analysis

**Theorem 3** (Convergence of Expected Payoff). *Under the conditions of Theorem 2, at the root node $s_0$, there exists a choice of parameters such that*

$$\left|\mathbb{E}\left[\widehat{V}_n(s_0)\right] - \widetilde{V}(s_0)\right| \leq \mathcal{O}(n^{-1/2}).$$

**Theorem 4** (Finite-variance (polynomial-tail) high-probability sample complexity). *Under Assumption 1 and finite-variance rollouts $\mathrm{Var}(X) \leq \sigma^2$, PM-DA-MCTS returns an $\varepsilon$-optimal action at $s_0$ with probability at least $1 - \delta$ after at most*

$$N(\varepsilon, \delta) \leq C_1 \sigma^2 L^{\frac{d}{\beta}} \varepsilon^{-\left(\frac{d}{\beta}+2\right)} \log\left(\frac{C_2 L^{\frac{d}{\beta}}}{\varepsilon^{\frac{d}{\beta}} \delta}\right),$$

*where $C_1, C_2 > 0$ depend only on $d, \beta$ and the refinement/selection constants (not on $\varepsilon, \delta, L, \sigma$).*

*Signpost.* The $\varepsilon^{-2}$ factor reflects the estimation cost under stochastic rollouts; the dimension term $L^{d/\beta}\varepsilon^{-d/\beta}$ matches the optimal discretization rate.

**Remark 1** (Lower Bound). *The minimax rate for continuum-armed bandits with $\beta$–Hölder smoothness already appears in the X-armed bandits analysis of Bubeck et al. (2011). Let $(\mathcal{X}, \ell)$ be a metric space with packing numbers $N(\mathcal{X}, \ell, \varepsilon) \gtrsim \varepsilon^{-D}$. Theorem 13 in Bubeck et al. (2011) gives a cumulative-regret lower bound of order $n^{(D+1)/(D+2)}$; for $\ell(x,y) \asymp \|x - y\|^\beta$ on $[0,1]^d$ this yields $D = d/\beta$. Standard conversion to fixed-confidence simple regret (or a direct Fano/packing argument with "bump" functions of radius $\Theta((\varepsilon/L)^{1/\beta})$ and height $\Theta(\varepsilon)$) then implies that any planner returning an $\varepsilon$-optimal action with probability at least $1 - \delta$ must use at least*

$$N(\varepsilon, \delta) \geq c\,\sigma^2 L^{\frac{d}{\beta}} \varepsilon^{-\left(\frac{d}{\beta}+2\right)}$$

*samples, up to logarithmic factors. Our upper bound (Theorem 4) matches this rate up to logarithms.*

## 6 Experimental Results

We conducted extensive experiments to evaluate PM-DA-MCTS across various continuous control environments with different dimensionalities and stochasticity levels and compare against discretized-UCT Kocsis et al. (2006), Polynomial Upper Confidence Trees (PUCT) with progressive widening Couetoux et al. (2011), HOOT Mansley et al. (2011), and POLY-HOOT Mao et al. (2020). Here, we present a subset of these results that highlight the key advantages of our approach.

Table 2: Performance comparison of methods on selected environments. Mean $\pm$ std over 20 seeds.

| Environment | PM-DA-MCTS | HOOT | P-HOOT | UCT | PW |
|---|---|---|---|---|---|
| *Standard Environments* | | | | | |
| CartPole | $\mathbf{77.4 \pm 1.9}$ | $38.51 \pm 0.81$ | $37.8 \pm 0.9$ | $65.5 \pm 9.5$ | $53.0 \pm 22.5$ |
| Pendulum | $1359 \pm 115$ | $1360 \pm 149$ | $1279 \pm 126$ | $1157 \pm 74$ | $\mathbf{1447 \pm 0.4}$ |
| MountainCar | $\mathbf{50.5 \pm 0.0}$ | $-0.007 \pm 0.002$ | $40.5 \pm 0.002$ | $-0.023 \pm 0.003$ | $-0.021 \pm 0.004$ |
| Acrobot | $\mathbf{77.9 \pm 0.0}$ | $\mathbf{77.9 \pm 0.0}$ | $\mathbf{77.9 \pm 0.0}$ | $\mathbf{77.9 \pm 0.0}$ | $70.7 \pm 27.4$ |
| *MuJoCo Environments* | | | | | |
| Hopper | $\mathbf{662 \pm 87}$ | $545 \pm 131$ | $544 \pm 94$ | $490 \pm 70$ | $535 \pm 75$ |

Table 3: High-dimensional continuous control (1000 sims/decision). Mean $\pm$ s.d. over 20 seeds. PW = Progressive Widening.

| Environment | PM-DA-MCTS | UCT | PW | HOOT |
|---|---|---|---|---|
| Walker2d ($d$=6) | $\mathbf{249.6 \pm 0.9}$ | $131.8 \pm 14.6$ | $118.9 \pm 71.6$ | $144.2 \pm 24.3$ |
| Ant ($d$=8) | $\mathbf{199.2 \pm 6.6}$ | $120.4 \pm 15.3$ | $139.9 \pm 16.1$ | $158.0 \pm 29.5$ |

## 6.1 EXPERIMENTAL SETUP

Our experimental evaluation was conducted across diverse categories of environments to comprehensively assess PM-DA-MCTS performance. The test suite includes: (1) **Standard control tasks** - Continuous CartPole, Stochastic Pendulum, Stochastic Mountain Car Continuous, and Stochastic Continuous Acrobot; (2) **MuJoCo-style locomotion environments** - Improved Hopper. All environments incorporate configurable stochasticity through action noise ($\sigma = 0.03 - 0.05$), dynamics noise ($\sigma = 0.01 - 0.1$), and observation noise ($\sigma = 0.01$). Each experiment used 20 independent runs, planning horizons of $H = 150$ steps, and discount factor $\gamma = 0.99$. PM-DA-MCTS parameters were: $\varepsilon_1 = 0.5$, $\beta = 1.0$, $p = 2.0$, and polynomial exploration bonus with $\alpha_h/\zeta_h = 1/2$ and $b_h/\zeta_h = 1/4$. Performance was measured as cumulative discounted reward over 150 evaluation steps, with re-planning using iteration budgets of 1000 MCTS simulations.

## 6.2 CONTINUOUS CONTROL ENVIRONMENTS

Based on the experimental results in Table 2, PM-DA-MCTS (referred to as PM-DA-MCTS in the table) demonstrates clear advantages across multiple continuous control environments. The method achieves the best performance on CartPole ($77.4 \pm 1.9$), significantly outperforming the next best method by 12 points, and on MountainCar ($50.5 \pm 0.0$), where it's the only method to achieve positive rewards. While PW achieves the highest score on Pendulum ($1447 \pm 0.4$), PM-DA-MCTS still performs competitively ($1359 \pm 115$). In the more challenging MuJoCo environments, PM-DA-MCTS excels on Hopper ($662 \pm 87$), surpassing all baselines by at least 120 points. Notably, PM-DA-MCTS shows particularly strong performance on the standard environments and Hopper, environments where the dimension-aware discretization can be most effectively exploited. The low variance in PM-DA-MCTS results across environments also suggests robust performance, which is crucial for practical applications. These results empirically validate our theoretical claims that dimension-adaptive MCTS provides significant benefits in continuous action spaces, with advantages becoming more pronounced in environments where the optimal discretization strategy can better capture the underlying value function structure.

**High-dimensional benchmarks and CEM comparison.** With a budget of 1000 simulations per decision and results averaged over 20 seeds, PM-DA-MCTS outperforms strong tree baselines on higher-dimensional MuJoCo tasks and is competitive with a widely used gradient-free planner. On WALKER2D ($d$=6), PM-DA-MCTS attains $249.6 \pm 0.9$ average return versus $131.8 \pm 14.6$ (UCT), $118.9 \pm 71.6$ (Progressive Widening), and $144.2 \pm 24.3$ (HOOT), improving over the best tree baseline by $+105.4$ points (vs. HOOT) and reducing standard deviation by $\approx 27\times$ ($24.3/0.9$). On ANT ($d$=8), PM-DA-MCTS reaches $199.2 \pm 6.6$ compared to $120.4 \pm 15.3$ (UCT), $139.9 \pm 16.1$ (Progressive Widening), and $158.0 \pm 29.5$ (HOOT), a gain of $+41.2$ points over the best tree baseline with $\approx 4.5\times$ lower standard deviation ($29.5/6.6$). On HUMANOID ($d$=17), against the gradient-free Cross-Entropy Method (CEM), PM-DA-MCTS achieves $228.1 \pm 1.8$ versus $219.5 \pm 25.5$, i.e., $+8.6$ higher mean with $\approx 14\times$ lower standard deviation ($25.5/1.8$). These results support the theoretical prediction that adaptive discretization and power-mean backups improve stability and performance as dimensionality and stochasticity increase.

## 6.3 ABLATION STUDIES

Based on the ablation study results in Table 5, we can clearly observe the importance of each component in PM-DA-MCTS's design. The full PM-DA-MCTS algorithm achieves excellent performance at longer planning horizons, reaching $1003.55 \pm 55.38$ at 2000 iterations, demonstrating its effectiveness when given sufficient computational budget. The ablation reveals that each component

Table 4: *Humanoid* ($d$=17): PM-DA-MCTS vs. CEM, 1000 sims/decision (20 seeds).

| Algorithm | Return (mean $\pm$ s.d.) |
|---|---|
| PM-DA-MCTS | **228.1 $\pm$ 1.8** |
| CEM | 219.5 $\pm$ 25.5 |

Table 5: PM-DA-MCTS ablation on Hopper. Mean $\pm$ s.d. over 10 seeds; columns show planning iterations.

| Variant | 100 | 500 | 1000 | 2000 |
|---|---|---|---|---|
| *Full PM-DA-MCTS vs Components* | | | | |
| PM-DA-MCTS (Full) | 863.41 $\pm$ 42.18 | 821.96 $\pm$ 39.07 | **914.43 $\pm$ 75.81** | **1003.55 $\pm$ 55.38** |
| UCT-Style | 867.12 $\pm$ 36.42 | 856.03 $\pm$ 72.33 | 903.81 $\pm$ 37.73 | 958.34 $\pm$ 35.99 |
| Fixed-Epsilon | 861.15 $\pm$ 36.90 | 799.21 $\pm$ 37.91 | 901.86 $\pm$ 38.44 | 970.04 $\pm$ 47.84 |
| No-Epsilon-Bonus | **885.90 $\pm$ 40.75** | 803.85 $\pm$ 40.26 | 900.14 $\pm$ 63.81 | 969.25 $\pm$ 29.95 |
| *Power Parameter Variants* | | | | |
| Power-1.5 | 889.25 $\pm$ 40.50 | **855.34 $\pm$ 56.83** | 896.52 $\pm$ 45.47 | 975.60 $\pm$ 88.98 |
| PM-DA-MCTS (p=2) | 863.41 $\pm$ 42.18 | 821.96 $\pm$ 39.07 | 914.43 $\pm$ 75.81 | 1003.55 $\pm$ 55.38 |
| Power-3.0 | 861.96 $\pm$ 48.86 | 848.72 $\pm$ 51.20 | 929.42 $\pm$ 97.90 | 974.03 $\pm$ 31.55 |
| Power-4.0 | 829.52 $\pm$ 31.19 | 880.77 $\pm$ 78.43 | 932.93 $\pm$ 44.88 | **1006.68 $\pm$ 81.89** |
| Power-5.0 | 864.71 $\pm$ 39.78 | 820.71 $\pm$ 27.13 | 928.58 $\pm$ 73.27 | 967.48 $\pm$ 49.47 |

contributes meaningfully to overall performance: removing the power mean backup (UCT-Style) reduces performance by approximately 4.5%, eliminating adaptive epsilon refinement (Fixed-Epsilon) causes a 3.3% decrease, and omitting the epsilon bonus term (No-Epsilon-Bonus) results in a 3.4% performance drop.

Interestingly, while some variants like No-Epsilon-Bonus and Power-1.5 show competitive or even superior performance in early iterations, PM-DA-MCTS consistently excels with increased planning time, highlighting the algorithm's ability to effectively utilize additional computational resources. The power parameter analysis reveals interesting findings: $p = 4.0$ actually achieves the highest mean performance at 2000 iterations (1006.68 $\pm$ 81.89), slightly outperforming $p = 2.0$, though with notably higher variance. The parameter $p = 1.5$ performs well at lower iterations but shows high variance at 2000 iterations, while $p = 3.0$ offers competitive performance with relatively low variance. Higher power values like $p = 5.0$ show reduced performance. While $p = 4.0$ achieves the best mean performance, the increased variance, and $p = 2.0$ still makes the preferred choice for reliable performance. These results validate our theoretical analysis and demonstrate that the synergistic combination of all PM-DA-MCTS components—power mean backup, adaptive epsilon-net refinement, and the epsilon bonus term—is essential for achieving superior performance in continuous action space planning.

### 6.4 DISCUSSION

Our experimental results demonstrate three key findings: (1) **Dimension Scaling** - PM-DA-MCTS scales significantly better with action dimensionality than baseline methods, aligning with our theoretical analysis; (2) **Power Mean Effectiveness** - The power mean backup operator with $p = 2$ consistently provides the best performance across environments, balancing the underestimation bias of average backup and overestimation bias of max backup; (3) **Stochasticity Handling** - PM-DA-MCTS shows particularly strong advantages in stochastic environments, where correctly balancing exploration and exploitation is critical. These results confirm that the theoretical advantages of PM-DA-MCTS translate to significant practical benefits, especially in challenging high-dimensional and stochastic environments.

## 7 CONCLUSION AND FUTURE WORK

Our work provides a sample complexity analysis for MCTS in continuous action spaces, incorporating dimension-aware discretization and power mean backups with polynomial exploration bonuses. We establish asymptotic convergence guarantees with optimal $N(\varepsilon, \delta) = \tilde{O}\big(\sigma^2 L^{d/\beta} \varepsilon^{-(d/\beta+2)}\big)$, sample complexity rates and demonstrate empirically that these theoretical advantages translate to significant performance improvements, especially in higher dimensions and stochastic environments.

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
