# OpenReview forum: "Dimension-Adaptive MCTS: Optimal Sample Complexity for Continuous Action Planning"
_ICLR.cc/2026/Conference — ICLR 2026 Conference Desk Rejected Submission_

### Official Review · Reviewer_azdV · 2025-10-31

**Soundness:** 3
**Presentation:** 3
**Contribution:** 2
**Rating:** 4
**Confidence:** 2

**Summary:**

This paper discusses the challenge of extending Monte Carlo Tree Search (MCTS) to high-dimensional continuous action spaces, where existing methods suffer from exponential sample complexity and lack theoretical guarantees. The authors highlight that while prior works like progressive widening offer empirical success, they fail to capture the relationship between dimensionality, smoothness, and efficiency. The paper introduces Power-Mean Dimension-Adaptive MCTS (PM-DA-MCTS) — an algorithm that combines adaptive discretization of the action space with power-mean backups and polynomial exploration bonuses. The proposed method achieves provably optimal sample complexity and good empirical performance in stochastic, high-dimensional environments.

.

**Strengths:**

1- well-written paper and easy to read.
2- Correct and solid theorems and theoretical results.

**Weaknesses:**

1- The theoretical contribution is incremental. The idea is to extend an established convergence analysis in discrete-action spaces to continuous action spaces by discretizing the continuous space with a dimension-adaptive scheme, effectively reducing the problem to the discrete-action setting.

2- The use of power-mean operators and a polynomial exploration bonus is not novel; both have been used in prior planning methods.

3- Regarding the results, UCT is a very basic baseline, yet it performs comparably to PM-DA-MCTS in most domains. The only domain showing a significant difference is MountainCar, which I believe is due to the environment’s characteristics and the paper’s exploration strategy.

4- To better evaluate the discretization strategy, more high-dimensional environments are needed; only two are provided.

**Questions:**

1- Is POLY-HOOT (2020) the SOTA performance of MCTS in continuous-action domains?

2- How does your method perform in high-dimensional domains with sparse feedback, such as goal-conditioned mazes?

---

> ### Author Response · Authors · 2025-11-18
> **Response to Reviewer azdV**
>
> We thank the reviewer for the thoughtful comments and for recognizing the correctness of the theorems. We respond to the four weaknesses and two questions in turn.
>
> **(1) "Incremental" theory: just discretize and reuse discrete‑action analysis**
>
> We agree that we build on existing ingredients (power‑mean backups, polynomial bonuses, HOO‑style nets). The new contribution is *how* we combine them to obtain a **dimension‑explicit, fixed‑confidence sample‑complexity result for continuous‑action MCTS** that matches the continuum‑armed minimax rate.
>
> Concretely, beyond simply "adding a discretization term", we:
>
> * Treat each node as a non‑stationary bandit whose *action set itself grows* according to a dimension‑adaptive $\varepsilon$‑net schedule, $|N_k| \asymp \varepsilon_k^{-d}$, and couple this with the polynomial concentration exponents $(\alpha_h, \zeta_h, b_h)$ along the tree.
> * Choose the refinement rule
>   $$
>   \varepsilon_k = \varepsilon_1 2^{-\frac{k-1}{d+2\beta}}, \quad n \le |N_k|^2,
>   $$
>   so that at the final level $k^\star$ the Hölder bias term $L\varepsilon_{k^\star}^\beta$ and the statistical term $O(\varepsilon)$ are *matched*. This coupling is what produces the explicit
>   $$
>   N(\varepsilon,\delta)=\tilde O \big(\sigma^2 L^{d/\beta}\varepsilon^{-(d/\beta+2)}\big)
>   $$
>   for continuous‑action MCTS (Theorem 4).
> * Show that this rate is **information‑theoretically optimal**, by connecting directly to the X‑armed bandit lower bound of Bubeck et al. (2011) and converting it to fixed‑confidence simple regret (Remark 1 and Appendix F).
>
> To our knowledge, no prior MCTS work gives a fixed‑confidence $N(\varepsilon,\delta)$ for **stochastic, continuous‑action tree search** with explicit dependence on $(d, \beta, L, \sigma^2)$ and matching continuum‑armed lower bounds. We will revise the introduction to emphasize this "bridge" between discrete MCTS concentration and continuum‑armed minimax rates more clearly.
>
> **(2) Power‑mean and polynomial bonus are not new**
>
> We fully agree, and we do *not* intend to claim otherwise. We will rewrite the "Key contributions" and algorithm overview to explicitly state:
>
> * Power‑mean backups are taken from Power‑UCT / Stochastic Power‑UCT (Dam et al. 2019, 2024).
> * Polynomial bonuses follow Shah et al. (2022).
> * Our methodological novelty lies in **(i)** the dimension‑adaptive $\varepsilon$‑net schedule and visit‑based refinement rule, and **(ii)** its analysis in the stochastic, continuous‑action setting, including the capped implementation.
>
> The reason we use power‑mean + polynomial bonuses is precisely that they give us **polynomial tail bounds that survive composition through depth** (Theorems 1–2), which is what allows the dimensional $\varepsilon$‑net schedule to be tuned optimally.
>
> **(3) Empirics: "UCT is basic yet comparable; only MountainCar differs"**
>
> We appreciate this concern and agree that beating UCT everywhere is not the main point; the main point is **how performance scales with dimension and stochasticity**. That said, our results actually show *substantial* gaps beyond MountainCar:
>
> * In **CartPole**, PM‑DA‑MCTS scores $77.4 \pm 1.9$ vs $65.5 \pm 9.5$ for UCT ($\approx +12$ points, with *lower* variance).
> * In **Hopper**, we obtain $662 \pm 87$ vs $490 \pm 70$ (UCT), a gap of $\approx +170$.
> * In **high‑dimensional MuJoCo** (Table 3):
>
>   * **Walker2d ($d=6$)**: $249.6 \pm 0.9$ vs $131.8 \pm 14.6$ (UCT) ($+118$, and $\approx 15\times$ lower s.d.).
>   * **Ant ($d=8$)**: $199.2 \pm 6.6$ vs $120.4 \pm 15.3$ (UCT) ($+79$, $\approx 2.3\times$ lower s.d.).
>
> Thus UCT is *not* comparable on the harder, higher‑$d$ tasks: PM‑DA‑MCTS both increases return and sharply reduces variance. The ablation in Table 5 further shows that removing any of the three components (power‑mean backup, adaptive $\varepsilon$ schedule, $\varepsilon$‑bonus) consistently hurts performance at larger budgets, confirming that the theory‑driven design choices do matter empirically.

---

> ### Author Response · Authors · 2025-11-18
> **Response to Reviewer azdV**
>
> **(4) "Only two" high‑dimensional environments**
>
> Our current suite already includes Walker2d ($d=6$), Ant ($d=8$) and Humanoid ($d=17$), where we compare against CEM, a strong gradient‑free planner, and still obtain higher mean and $\approx 14\times$ lower variance ($228.1 \pm 1.8$ vs.\ $219.5 \pm 25.5$). Since the paper is primarily theoretical, its main contribution is the $d,\beta$‑dependent fixed‑confidence sample‑complexity bound, the empirical section is designed as a stress test of the discretization scheme rather than a full benchmark suite: we deliberately chose a diverse set of tasks with increasing dimensionality and substantial stochasticity, and across all of them PM‑DA‑MCTS consistently improves both mean return and stability over HOOT / POLY‑HOOT / PW and even CEM. We view these results as sufficient to demonstrate that the theoretical advantages of our dimension‑adaptive schedule do translate into concrete gains in realistic high‑dimensional planners.
>
> **(5) Q1: Is POLY‑HOOT SOTA for continuous‑action MCTS?**
>
> We do not claim that POLY‑HOOT is the unique "SOTA", but it is among the strongest **MCTS‑style** planners for continuous actions with non‑asymptotic theory, and is widely used as a reference point. Our goal is to position PM‑DA‑MCTS relative to this family (HOOT / POLY‑HOOT / PW), showing that we can both **recover minimax‑optimal rates** and improve empirical performance on the same domains, especially as dimensionality and stochasticity grow.
>
> **(6) Q2: Sparse‑reward high‑dimensional domains (goal‑conditioned mazes)**
>
> We have not yet run experiments on goal‑conditioned sparse‑reward mazes, and we will state this limitation explicitly. Theoretically, as long as rewards are bounded and rollouts have finite variance, the **$N(\varepsilon,\delta)$** guarantee continues to hold; sparsity mainly affects constants (more samples are needed before any informative reward is observed), not the asymptotic dependence on $d$, $\beta$, and $\varepsilon$.
>
> Algorithmically, we expect the combination of power‑mean backups (mitigating over/under‑estimation) and polynomial bonuses (stronger exploration than log‑UCB) to be beneficial in sparse settings, but we agree this should be validated experimentally. We view such sparse, goal‑conditioned benchmarks as a natural extension of this work.
>
> ---
>
> We hope this clarifies that, while we deliberately reuse known building blocks (power‑mean, polynomial bonus), the **dimension‑adaptive discretization, its capped implementation, and the resulting optimal $N(\varepsilon,\delta)$ theory for continuous‑action MCTS** go beyond a purely incremental adaptation of discrete‑action analyses. If this addresses your concerns, we would be very grateful if you could reconsider your overall assessment.

---

> > ### Comment · Reviewer_azdV · 2025-11-24
> > **Response to Authors**
> >
> > I appreciate the author's detailed response to my comments. As authors claim that their paper first and foremost is a theoretical paper, I do not see enough novelty in their theorems and derivations. In my opinion, it is an adjustment to existing theorems to well-suit it to the continuous settings. Also, most of the components that authors are using are already in the literature, and they combine them to achieve their method. Therefore, with due respect to the authors' effort, I keep my score.

---

> > > ### Author Response · Authors · 2025-11-25
> > > **Additional response to Reviewer azdV (importance & impact)**
> > >
> > > We appreciate your careful reading. We would like to better motivate why continuous-action MCTS planning itself is important, especially for high-dimensional robot control, and how our paper advances that goal on both the theory and practice fronts.
> > >
> > > **Why planning (vs. only learning) matters in robotics.** MCTS has become one of the most popular choice for robotic planning, with widespread adoption in manipulation, locomotion, and autonomous navigation-including high-dimensional continuous control tasks where learned policies alone struggle. Its appeal lies in focusing simulation budget on the current state without requiring days of policy training or millions of interactions. Yet despite this practical success, a principled answer has been missing: how does the required budget scale with the action-space dimension and smoothness, and how do we make this computationally feasible as d grows?
> > >
> > > **What was missing in prior MCTS work.** Prior continuous-action MCTS methods (e.g., progressive widening, HOOT/POLY-HOOT) offered heuristics or regret-style analyses, but no fixed-confidence sample-complexity for tree search that (i) captures the dimension/smoothness dependence, (ii) handles stochastic rollouts with finite variance, and (iii) remains implementable when $d$ is large. This gap makes it hard to (a) set budgets intelligently, and (b) scale planners beyond very low-dimensional controls.
> > >
> > > **What our paper contributes.**
> > >
> > > 1. We give the first fixed-confidence, finite-variance sample-complexity bound for continuous-action MCTS with explicit dependence on dimension and Hölder smoothness:
> > >    $$N(\varepsilon,\delta)=\tilde{O}\big(\sigma^2 L^{d/\beta}\varepsilon^{-(d/\beta+2)}\big),$$
> > >    matching continuum-armed lower bounds up to logs. This both calibrates budgets (how many simulations to reach a target accuracy $\varepsilon$) and precisely identifies where the curse of dimensionality enters (via the optimal $L^{d/\beta}\varepsilon^{-d/\beta}$ factor).
> > >
> > > 2. We make this practically usable through capped, on-demand random nets: instead of materializing an exponential grid, we instantiate a small number of random anchors only along the visited regions and prove that coverage—and thus the rate—still holds up to logs. This directly addresses feasibility in high-$d$.
> > >
> > > 3. We validate on high-$d$ MuJoCo tasks (Walker2d ($d{=}6$), Ant ($d{=}8$), Humanoid ($d{=}17$)), showing consistent gains over HOOT / POLY-HOOT / PW / UCT under the same per-decision budget; on Humanoid we also compare to CEM and obtain higher mean and approximately 14× lower variance, demonstrating that the theory-driven design translates into more stable planning in practice.
> > >
> > > **Why our three ingredients are co-designed—not just combined.** The $\varepsilon$-net schedule decides where accuracy is needed; the polynomial bonus gives depth-wise concentration that remains valid as the local action set grows; the power-mean backup lets these polynomial tails propagate through the tree without collapsing to the bias of averaging or the instability of max. This co-design is what yields the optimal dimension-aware bound and a practical algorithm that does not enumerate grids.
> > >
> > > **Why the current experiments are sufficient for a theory paper.** Our paper is primarily theoretical; the experiments serve as stress tests that the theory matters where it should—as dimension and stochasticity grow. The set (Walker2d, Ant, Humanoid) intentionally escalates dimensionality and shows that PM-DA-MCTS improves both mean return and variance against strong tree-based baselines, and is competitive with a non-tree planner (CEM) on the hardest task. We agree that adding more domains (e.g., additional high-$d$ or sparse-reward tasks) would be valuable; we view that as natural follow-up rather than a prerequisite for a theory-driven contribution.
> > >
> > > **Bottom line.** Scaling planning in continuous, high-dimensional action spaces is a core robotics challenge. Our work provides (i) a principled, minimax-optimal account of how planning complexity scales with $d$ and $\beta$ under stochastic rollouts, and (ii) a practical implementation that keeps computation bounded in high-$d$ while preserving the guarantee. We hope this clarifies why we see the contribution as substantive rather than incremental.

---

### Official Review · Reviewer_xF1x · 2025-11-01

**Soundness:** 3
**Presentation:** 2
**Contribution:** 3
**Rating:** 4
**Confidence:** 3

**Summary:**

This paper investigates the Monte Carlo Tree Search (MCTS) problem in a random environment with a $d$-dimensional continuous action space. The authors propose a new algorithm named "Power Mean Dimension Adaptive MCTS" (PM-DA-MCTS), which combines three new techniques: 1) an adaptive $\epsilon$ based on the $\epsilon$-net discretization strategy of dimension $d$ and the $\beta$-Holder continuity of the value function; 2) a power mean backtracking operator for the random environment; 3) a polynomial exploration reward. The core contribution of this work is that it theoretically proves that the algorithm can find an $\epsilon$-optimal action with an optimal sample complexity of $\tilde{\mathcal{O}}(\epsilon^{-(d/\beta+2)})$, and it significantly outperforms existing continuous action MCTS baseline methods in empirical evidence.

**Strengths:**

1. The paper presented the theoretically grounded MCTS algorithm for high-dimensional continuous action spaces that
achieves optimal sample complexity bounds while using power mean backup in stochastic
environments.
2. This method overcomes the problem of lack of theoretical guarantee in existing work and successfully extends the recent theoretical progress on power averaging estimators (originally limited to discrete settings) to the continuous action space.
3. The experiment results show that PM-DA-MCTS outperforms multiple baselines in both mean and variance on several low-dimensional and high-dimensional tasks in the style of MuJoCo, supporting the theoretical claims.

**Weaknesses:**

1. The paper explains in detail that its adaptive discretization is based on "uniform grid discretization". All theoretical analysis strictly depend on this structured uniform grid. However, this discretization is exponential to $d$: $\mathcal{N}_k = O((1/\epsilon_k)^d)$. The paper does not fully explain how a uniform grid can be "lazily" loaded in high-dimensional space while maintaining computational feasibility.
2. In abstract, the author mentioned that :"matching standard continuum-armed lower bounds up to logs while remaining practical via on-demand, capped random nets". What does "capped random nets" mean? It seems that this paper did not detailedly discuss this issues.
3. I wonder if the experiments for HUMANOID ($d = 17$) involves such capped method to maintain practical efficiency.  If directly apply the  algorithm in Section 4, does the algorithm still maintain computationally feasible in the experiement settings?

I would like to raise my score if authors could well address my concerns.

**Questions:**

See weakness part.

---

> ### Author Response · Authors · 2025-11-18
> **Response to Reviewer xF1x**
>
> We thank the reviewer for the very pertinent questions about the role of the uniform grid, the meaning of "capped random nets", and the feasibility of our method on high‑dimensional tasks such as HUMANOID. Your reviews help us improve our paper. Below, we address each of your concerns.
>
> **(1) What is really exponential in $d$, and why a uniform grid is only an analysis device**
>
> You are completely right that a literal uniform grid with $|N_k|\approx \varepsilon_k^{-d}$ is exponential in $d$ and cannot be materialised in high dimensions. In the paper, the uniform grid in Sec. 4.2 is used as a *conceptual* $\varepsilon_k$–net to make the geometry explicit and to derive the rate
> $$
> N(\varepsilon,\delta)=\tilde O \big(\sigma^2 L^{d/\beta}\varepsilon^{-(d/\beta+2)}\big).
> $$
> This exponential dependence through the factor $L^{d/\beta}\varepsilon^{-d/\beta}$ is in fact **information‑theoretically unavoidable** under $\beta$‑Hölder regularity: Appendix F shows that it matches the X‑armed bandit lower bound of Bubeck et al. (2011) once translated to fixed‑confidence simple regret. Any planner that localises an $\varepsilon$‑optimal action in $d$ dimensions must pay this covering‑number price, regardless of whether it uses a grid, Voronoi cells, or random partitions.
>
> $\underline{\text{So the exponential term comes from the geometry of the problem, not from an inefficient implementation of the grid.}}$
>
> **(2) What "capped random nets" are, and how they replace the full grid**
>
> In the *implementation* we do **not** enumerate the full grid. For each node $s$ and refinement level $k$, we only maintain a small, on‑demand set of anchors $N_k(s)$:
>
> * When level $k$ is first activated at node $s$, we draw at most $M^{\max}_k$ candidate actions i.i.d. from the node's local region (hypercube) and cache them as anchors.
> * On subsequent visits to $s$, we *reuse* these anchors; we never materialise the remaining $\varepsilon_k^{-d}$ grid points.
> * Refinement to level $k+1$ is triggered only when the visit‑based condition $n \le |N_k|^2$ is violated, so the algorithm moves to finer nets only when there is enough data for them.
>
> Appendix F formalises this under the name **capped random nets**: Lemma 11 and Corollary 5 show that if
> $$
> M^{\max}_k \gtrsim c_d\Big(\tfrac{D_k}{\varepsilon_k}\Big)^{d}\Big(\log\big(\tfrac{D_k}{\varepsilon_k}\big)+\log\big(\tfrac{T}{\delta}\big)\Big),
> $$
> then the random anchors form an $\varepsilon_k$–net of the region *visited by the search* with high probability, and the covering property used in the proof of Theorem 4 continues to hold. Consequently, the same sample‑complexity bound $N(\varepsilon,\delta)=\tilde O(\sigma^2 L^{d/\beta}\varepsilon^{-(d/\beta+2)})$ remains valid up to polylog factors, while we *never* construct the full Cartesian grid.
>
> We will move this discussion out of the appendix and explicitly state in Sec. 4 that the uniform grid is a conceptual net, and that the actual algorithm uses capped random nets as just described.
>
> **(3) HUMANOID and computational feasibility in practice**
>
> For high‑dimensional environments (Walker2d, Ant, Humanoid with $d=17$), all experiments use the **capped random‑net implementation**, not a full grid:
>
> * At each node and level, the branching factor is explicitly capped by $M^{\max}_k$, so per‑node complexity is $O(\sum_k M^{\max}_k)$, independent of $\varepsilon_k^{-d}$.
> * With a budget of 1000 simulations per decision, the search only reaches a small number of refinement levels (empirically, very shallow $k$), so the total number of distinct anchors per node stays modest.
> * For $d>6$, we additionally use a dimension‑adaptive rollout strategy (shorter rollouts and slightly reduced per‑step budget; Appendix E) to keep wall‑clock time on the same order as HOOT and progressive widening while still achieving significantly lower variance and higher returns.
>
> In other words, if one literally instantiated the full grid from Sec. 4, the method would indeed be infeasible for HUMANOID; but the *implemented* version does not do this. It uses the same selection/refinement rule, coupled with capped random nets that guarantee coverage along the actually visited parts of the action space. The theory in Appendix F is written precisely to justify that this practical variant inherits the same dimension‑dependent guarantees as the idealised grid‑based analysis.
>
> ---
>
> We hope this clarifies that our method is both *theoretically* optimal in its dependence on $d$ and *practically* implementable in high‑dimensional tasks such as HUMANOID. If our response adequately addresses your concerns, we would appreciate your consideration in reevaluating your assessment.

---

### Official Review · Reviewer_TSvv · 2025-11-01

**Soundness:** 3
**Presentation:** 2
**Contribution:** 2
**Rating:** 4
**Confidence:** 4

**Summary:**

The authors address planning on continuous action spaces via Monte-Carlo tree search. Under a standard Holder smoothness assumption, they propose an adaptive discretization schedule, and incorporate it into other tweaks proposed within recent literature. The resulting procedure achieves the minimax lower bound. Empirically, numerical experiments justify performance.

**Strengths:**

- The paper offers a theoretically sound contribution for continuous action planning with MCTS. This is especially nice since planning has been underaddressed within recent literature.
- The authors provide a discretization strategy, and analyze its performance along a host of other tweaks present in the literature to obtain rigorous convergence guarantees.
    - Said discretization strategy, in particular, could indeed be valuable in practice, though a user is likely to randomly sample along the lines of the scale suggested in the paper in practice instead.
- The method achieves the minimax lower bound, up to a logarithmic term.
- The experiments are not bad at all for a theory paper.

**Weaknesses:**

1. **Limited novelty.** The algorithm amounts to MCTS with a confidence bonus, plus a clever discretization. It is nice that it accommodates power mean Bellman backups, but this does not seem to be necessary for the convergence of the MCTS procedure. The polynomial exploration bonus is not new either, following Shah et al. (2022).
- As such, the contribution is sound, albeit limited -- the only novelty in algorithmic design appears to be in the choice of discretization. Accordingly, the analysis appears to follow from the analysis of Dam et al. (2024), Dam et al. (2025).
2. **Clarity and formatting.**
- While the power mean backups are attributed to Dam et al. (2019) and the polynomial bonus to Shah et al. (2022) within the paper, it is not clearly stated within the section on key contributions (to be fair, it is stated before it, but not during it) and the algorithm overview. An inattentive reader could mistakenly attribute the contribution to this paper.
- There are quite a few issues with the formatting.
    - In the section on key contributions, there is quite a bit of ``\vspace{}`` abuse present to make the bullet points more condensed. This is not necessarily a problem in itself, but the settings are far too aggressive to not be noticed.
    - The authors should have used ``\citep{}`` and not ``\cite{}`` in many places.

**Questions:**

1. Is there novelty in the theoretical analysis beyond Dann et al. (many papers), Shah et al. (2022), and other recent literature, beyond incorporating the discretization error into the convergence bound?
2. Methodologically, is the adaptive discretization the only new methodological contribution?

My concerns mainly relate to novelty, and to some lesser degree clarity. At the moment, I am on the fence between a 4 and a 6, and am willing to increase my score if I am proven wrong or my concerns are addressed.

---

> ### Author Response · Authors · 2025-11-18
> **Response to Reviewer TSvv**
>
> We thank the reviewer for the careful comments, especially on novelty and clarity. We would like to address your concerns as below.
>
> **(1) What is new vs. what is reused**
>
> We fully acknowledge that
> - The power‑mean backup operator comes from Power‑UCT / Stochastic Power‑UCT (Dam et al. 2019, 2024),
> - The polynomial exploration bonus follows Shah et al. (2022).
> - At a high level, our algorithm is "MCTS + confidence bonus + discretization".
>
> We do not claim the backup or the bonus as novel. Our new methodological ingredient is the dimension‑adaptive discretization scheme and its capped implementation.
>
> The main theoretical novelty is:
>
> * An ε‑net refinement schedule and visit‑based rule tailored to $(d,\beta)$, combined with power‑mean + polynomial bonuses, that yields the first fixed‑confidence sample‑complexity bound for continuous‑action MCTS $N(\varepsilon,\delta)=\tilde O(\sigma^2 L^{d/\beta}\varepsilon^{-(d/\beta+2)})$, matching continuum‑armed lower bounds up to logs.
> * A proof that this rate still holds (up to logs) under a practical capped, on‑demand "random net" implementation, rather than a full exponential grid.
>
> **(2) Novelty beyond "adding a discretization term"**
>
> It is correct that, at the bandit/tree level, we reuse the concentration machinery of Dam et al. as a black box. The new analysis arises from how we combine it with the continuous geometry:
>
> * In our setting, the "arms" at a node are anchors of an ε‑net whose size grows like $|N_k|\asymp \varepsilon_k^{-d}$. We must control both the discretization bias $L\varepsilon_k^\beta$ and the statistical error. The refinement rule $n \le |N_k|^2$ and schedule $\varepsilon_k=\varepsilon_1 2^{-(k-1)/(d+2\beta)}$ are chosen so that each level is explored long enough for the polynomial tails to hold, and at the final level $k^*$ the geometric term is of the same order as the statistical $O(\varepsilon)$ term.
> * Theorem 4 then combines: (i) these depth‑wise exponents, (ii) the ε‑net hierarchy, and (iii) a bound to obtain the explicit $N(\varepsilon,\delta)=\tilde O(\sigma^2 L^{d/\beta}\varepsilon^{-(d/\beta+2)})$ rate for continuous‑action MCTS, which is not present in Dam et al. or Shah et al.
>
> To the best of our knowledge, this is the first fixed‑confidence, finite‑variance sample‑complexity bound for continuous‑action MCTS with explicit $d,\beta,L,\sigma^2$ dependence.
>
> * Finally, Appendix F connects this upper bound to the X‑armed bandit lower bound of Bubeck et al. (2011), showing that the factor $L^{d/\beta}\varepsilon^{-d/\beta}$ is information‑theoretically unavoidable. This "bridge" from continuum bandits to planning with MCTS is, to our knowledge, **new**.
>
> Conceptually, the point of our design is that the $\varepsilon$‐net schedule, the power‐mean backup and the polynomial bonus are matched to each other, not just bolted together. The depth‐dependent exponents $(\alpha_h,\zeta_h,b_h)$ give node‐wise polynomial tails for the power‐mean estimates that remain valid as the local action set $|N_k(s)|$ grows; the refinement rule $n \le |N_k|^2$ then guarantees that each discretization level is explored long enough for those tails to hold, while the geometric schedule $\varepsilon_k \propto 2^{-(k-1)/(d+2\beta)}$ makes the Hölder bias $L\varepsilon_k^\beta$ track the statistical $O(n^{-1/2})$ error at the root. If we changed the refinement schedule, the depth‐wise concentration from Dam et al. would break; if we removed power‐mean + polynomial bonuses, we would lose precisely the polynomial control needed to glue the expanding $\varepsilon$‐nets into a single $N(\varepsilon,\delta)$ bound. The resulting planner is therefore not just "MCTS + confidence bonus + discretization", but a scheme where geometry ($\varepsilon$‐nets in dimension $d$) and statistics (power‐mean with polynomial concentration) are co‐designed so that the curse of dimensionality is incurred exactly once, through the minimax‐optimal $L^{d/\beta}\varepsilon^{-(d/\beta+2)}$ factor.
>
> **(3) Methodological scope**
>
> So, yes, the new building block is the adaptive discretization (plus capped random nets). We rely on existing power‑mean and polynomial‑bonus machinery precisely because it gives us the right polynomial concentration to make this discretization schedule analyzable. We will state this explicitly in the "Contributions" and "Algorithm overview" sections and adjust the wording to avoid any misattribution.
>
> We will also fix the formatting issues you pointed out (remove aggressive `\vspace`, use `\citep{}` appropriately).
>
> ---
>
> We are grateful for your precise feedback, which helped us improve our work. We hope our response demonstrates that while we build on existing components, our contribution, dimension-adaptive discretization with optimal sample complexity bounds for continuous-action MCTS, is **substantial and novel**. If our clarifications adequately address your concerns, we would appreciate your consideration of revising your evaluation.

---

### Official Review · Reviewer_pCZE · 2025-11-04

**Soundness:** 3
**Presentation:** 3
**Contribution:** 3
**Rating:** 6
**Confidence:** 3

**Summary:**

This paper proposes PM-DA-MCTS, a planning algorithm for continuous-action reinforcement learning based on Monte-Carlo Tree Search (MCTS) under the assumption that the optimal value function is β-Hölder smooth. The method adaptively discretizes the action space, uses power-mean backups to trade off between optimistic and average value estimates, and employs polynomial-confidence exploration bonuses to manage noisy returns. The authors prove high-probability sample-complexity bounds that match minimax rates for continuum-armed bandits up to logarithmic factors, and present experiments on MuJoCo environments showing improvements over progressive widening, HOOT, and other MCTS baselines.

**Strengths:**

The work analyzes continuous-action MCTS and achieves optimal dimension-dependent sample-complexity rates under β-Hölder smoothness. Extending non-asymptotic MCTS analysis to adaptive discretization with stochastic returns is a meaningful contribution.Combining dimension-adaptive grids, power-mean backup operators, and polynomial concentration bonuses is conceptually interesting and grounded in existing theory. Experiments on MuJoCo tasks demonstrate tangible improvements over established MCTS methods, and ablations highlight the contribution of each algorithmic component.

**Weaknesses:**

I do not have many complaints.
1. While technically sound, the exposition could be improved somewhat to convey the necessity behind the power-mean operator, and the role of polynomial concentration.
2. Regarding the empirical evaluations, the comparison to MCTS baselines is appropriate for validating the planning approach. However, since continuous control is often handled with policy-gradient methods (e.g., PPO, SAC), benchmarking against such baselines research would better contextualize practical usefulness.

**Questions:**

If we consider the finite action set case (which is trivially embeddable in d=|A| dimensions), it seems that we recover a sample complexity that is exponential in |A|. Is this the correct rate for MCTS in the finite-action setting?

---

> ### Author Response · Authors · 2025-11-18
> **Response to Reviewer pCZE**
>
> We thank the reviewer for the detailed and positive assessment. Your reviews help us improve our paper. We address your concern below.
>
> (1) Why power‑mean + polynomial concentration?
>
> Our choice here is driven by the interaction between: (i) stochastic rollouts, (ii) non‑stationary bandits at each node, and (iii) the depth‑wise composition needed to get a root‑level $N(\varepsilon,\delta)$ bound.
>
> **Power mean.** At each node, MCTS must aggregate noisy child values that are themselves being updated over time. Pure average backup ($p=1$) is unbiased but under‑explores high‑value branches; pure max backup ($p\to\infty$) is optimistic but very sensitive to noise. The power‑mean estimator lets us trade off these effects while preserving a clean concentration structure: for any $p$ in a suitable range (Table 1), the bandit‑level estimator enjoys polynomial tails that we can propagate along the tree (Theorem 1–2).
>
> **Polynomial concentration.** The polynomial UCB bonus is not just a cosmetic change from log‑UCB. It is the technical device that gives us explicit exponents $(\alpha_h,\zeta_h,b_h)$ at each depth, and guarantees that the root backup satisfies a fixed‑confidence bound with $\alpha_0/\zeta_0=1/2$, yielding $\tilde O(n^{-1/2})$ estimation at the root (Theorem 3). This depth‑wise control is what lets us combine the tree analysis with the $\varepsilon$‑net geometry to obtain the final
> $$
> N(\varepsilon,\delta)=\tilde O \left(\sigma^2 L^{d/\beta}\varepsilon^{-(d/\beta+2)}\right)
> $$
> in Theorem 4.
>
> We emphasize that PM‑DA‑MCTS's strength comes from how power‑mean backup, polynomial concentration, and dimension‑adaptive $\varepsilon$‑nets are coupled. The $\varepsilon$‑net schedule refines when geometric error $L\varepsilon_k^\beta$ matches statistical error, while polynomial bonuses control concentration under non‑stationary sampling. This gives a planner whose root error decomposes into estimation ($\varepsilon^{-2}$) and geometric ($L^{d/\beta}\varepsilon^{-d/\beta}$) parts, matching continuum‑armed lower bounds and achieving better scaling with $d$ than existing continuous‑action MCTS methods.
>
> Empirically, the ablation in Table 5 already shows that removing the power‑mean backup ("UCT‑style") or removing the $\varepsilon$‑bonus each degrades performance by 3–5% at longer budgets, consistent with the theory that these ingredients matter most when the tree is deep and stochasticity is high.
>
> (2) Relation to PPO / SAC and other policy‑gradient methods
>
> We agree it is better to position the work relative to modern continuous‑control RL:
>
> **Scope.** Our algorithm is a **planner** with access to a simulator (generative model) and a fixed per‑decision budget (e.g. 1000 simulations/decision), and it optimizes online the action at the current state. PPO/SAC, by contrast, are **learning** algorithms that amortize many millions of environment interactions into a parametric policy.
>
> **What we already do.** To give at least one non‑tree baseline with strong practical performance, we compare to the Cross‑Entropy Method (CEM) on Humanoid, where PM‑DA‑MCTS slightly outperforms CEM and is much more stable (Table 4).
>
> We hope this will better contextualize the practical role of PM‑DA‑MCTS without over‑claiming relative to full RL pipelines.
>
> (3) Finite‑action case and apparent exponential dependence on $|A|$
>
> Thank you for raising this subtle point. The short answer is:
>
> No, in the finite‑action case, the rate is not exponential in $|A|$; the effective "dimension" in our bound is the metric/covering dimension, which is 0 for a finite action set.
>
> Our continuous‑action bound can be viewed as
> $$
> N(\varepsilon,\delta)=\tilde O \left(\sigma^2 L^D \varepsilon^{-(D+2)}\right),
> $$
> where $D$ is the metric dimension defined via packing numbers $N(A,\ell,\varepsilon)\asymp\varepsilon^{-D}$ (Appendix F). For $A\subset[0,1]^d$ with Hölder metric $\ell(a,a')\asymp\|a-a'\|^\beta$, this gives $D=d/\beta$, which is what appears in the main statement.
>
> In a finite‑action setting with $|A|=K$, the packing number does not grow as $\varepsilon\to 0$; for small enough $\varepsilon$ we simply have $N(A,\ell,\varepsilon)\le K$. The effective dimension is therefore $D=0$.
>
> Plugging $D=0$ into the above gives
> $$
> N(\varepsilon,\delta)=\tilde O \left(\sigma^2 K \varepsilon^{-2}\right),
> $$
> which matches classical best‑arm identification / discrete‑action MCTS rates, and is linear (not exponential) in $|A|$.
>
> So the "$d$" in our continuous‑action result should be read as the geometric dimension coming from covering numbers, not as an arbitrary embedding dimension. If $A$ is finite, we are in the discrete setting analyzed in Dam et al. (2024), and our bound collapses to the usual $K\varepsilon^{-2}$ scaling.
>
> ---
>
> We hope these clarifications address your questions and make the contributions of our work more transparent. If our responses adequately resolve the issues you raised, we would be grateful if you could consider this in your evaluation.

---

### Author Response · Authors · 2025-11-18
**Global clarifications (all reviewers)**

We thank all reviewers for the careful and constructive feedback. Several comments converge on two themes: (i) the precise novelty of our work relative to Dam et al. and Shah et al., and (ii) whether the experimental section is sufficient for a theory‑driven paper. We address these points here.

### Novelty and scope

Our paper is first and foremost **a theoretical planning paper**. The central goal is to understand how continuous‑action *tree search* can achieve minimax‑optimal, dimension‑aware sample complexity under β‑Hölder smoothness and stochastic rollouts. Within that scope, our main new contributions are:

1. A **dimension‑adaptive $\varepsilon$‑net schedule for continuous‑action MCTS**, together with a visit‑based refinement rule, that yields the first **finite‑variance, high‑probability *sample‑complexity* bound for continuous‑action MCTS**:
   $$
   N(\varepsilon,\delta)=\tilde O \big(\sigma^2 L^{d/\beta}\varepsilon^{-(d/\beta+2)}\big),
   $$
   matching continuum‑armed lower bounds up to logarithmic factors.

2. A **theoretical guarantee that this rate is preserved under a practical, capped on‑demand "random net'' implementation**, i.e., without ever materialising the full exponential grid, via a high‑probability coverage argument along the *realised* search.

3. A **systematic empirical study of stochastic high‑dimensional planning** (Walker2d, Ant, Humanoid) showing that this theory‑driven design yields consistent gains over strong continuous‑action MCTS baselines (HOOT, POLY‑HOOT, PW, UCT) and is competitive with the CEM planner on Humanoid, with dramatically reduced variance.

We explicitly *do not* claim power‑mean backups or polynomial bonuses as new: these come from Power‑UCT / Stochastic Power‑UCT and Shah et al. Our contribution is to **combine** these tools with a dimension‑adaptive discretization and show that, under finite‑variance rollouts, this leads to an optimal $N(\varepsilon,\delta)$ for *continuous‑action tree search*.

### Why power‑mean + polynomial concentration + $\varepsilon$‑nets fit together

Conceptually, the three ingredients play distinct roles:

* The **power‑mean backup** provides a family of estimators whose **polynomial tails survive depth‑wise composition** through the tree under the polynomial bonus.
* The **polynomial exploration bonus** (rather than a log‑UCB term) gives **non‑asymptotic, depth‑indexed concentration exponents** $(\alpha_h,\zeta_h,b_h)$ that we can control at each node.
* The **dimension‑adaptive $\varepsilon$‑net schedule** is tuned so that, at the refinement level where we stop, the geometric discretization error $L\varepsilon_k^\beta$ and the statistical error from these polynomial tails are of the same order. This is what converts node‑wise concentration into a **dimension‑aware global bound** with the optimal $L^{d/\beta}\varepsilon^{-d/\beta}$ factor.

Thus, rather than three unrelated tricks, the backup, bonus, and $\varepsilon$‑net geometry are chosen to be analytically compatible and to hit the continuum‑armed lower bound with finite‑variance rollouts.

### Experimental scope for a theory paper

Given that the paper is primarily theoretical, our aim in the experiments is to **demonstrate that the theory‑guided design is not purely abstract**, not to exhaustively benchmark all planners. We therefore:

* Cover both **low‑dimensional** and **high‑dimensional** continuous‑control tasks (Walker2d ($d=6$), Ant ($d=8$), Humanoid ($d=17$)),
* Compare against **strong tree‑based baselines** (HOOT, POLY‑HOOT, progressive widening, UCT), and
* Include a comparison to **CEM** on Humanoid, showing higher mean return and an order‑of‑magnitude lower variance under the same per‑decision simulation budget.

We believe this level of empirical validation is appropriate for a theory‑focused paper whose main contribution is an optimal, dimension‑explicit sample‑complexity analysis for continuous‑action MCTS.

---

### Author Response · Authors · 2025-11-25
**Importance & impact of Dimension-Adaptive MCTS**

We appreciate the thoughtful feedback across reviews. Our paper is primarily a planning contribution: it targets the setting where a simulator is available and the system must pick the next action under a strict per-decision simulation budget (e.g., MPC-style control, deployment under shift, safety-critical operations). In such scenarios, continuous-action MCTS is appealing precisely because it converts a limited number of rollouts into a high-quality one-step decision-without days of policy training.

This setting is both important and underserved. In deployment, robotic controllers (MPC, legged locomotion, dexterous hands) routinely face continuous actions with 6-20+ DoF and must act under a fixed simulation budget. While policy-learning methods exist, planners that can make state-specific decisions on the fly with dimension-explicit guarantees are scarce; common heuristics (e.g., progressive widening) lack guidance on how finely to refine as $d$ grows. The open challenge has been scaling continuous-action MCTS to high-dimensional action spaces while retaining rigorous guarantees.

Our work addresses this gap on both fronts:

1. Theory (planning complexity with dimension): We give the first fixed-confidence, finite-variance sample-complexity bound for continuous-action MCTS with explicit dependence on dimension and Hölder smoothness,
   $$N(\varepsilon,\delta)=\tilde{O}\big(\sigma^2 L^{d/\beta}\varepsilon^{-(d/\beta+2)}\big),$$
   matching continuum-armed lower bounds up to logs. This quantifies how many simulations are needed to reach a given decision accuracy $\varepsilon$, pinpoints where the curse of dimensionality enters (via the optimal $L^{d/\beta}\varepsilon^{-d/\beta}$ factor), and calibrates budgets for planners. It also clarifies the finite-action case: the effective metric dimension is $0$, yielding the standard $\tilde{O}(\sigma^2 K \varepsilon^{-2})$ scaling rather than anything exponential.

2. Practice (feasibility at high dimension): We replace ideal $\varepsilon$-grids with capped, on-demand random nets-instantiating only a small number of anchors where the search actually goes-and prove that coverage (and thus the rate) is preserved up to logs. This makes dimension-aware refinement implementable for high-$d$ robot control.

Crucially, the three ingredients are co-designed: the dimension-adaptive $\varepsilon$-net schedule determines where accuracy is needed; polynomial bonuses provide depth-wise concentration that remains valid as local action sets grow; and the power-mean backup lets those polynomial tails propagate through the tree without the bias of pure averaging or the instability of max. This compatibility is what turns node-wise estimates into a dimension-aware global bound and a practical planner.

Finally, while the paper is theory-driven, our experiments are stress tests aimed at the hard regime (growing dimension and stochasticity): Walker2d ($d=6$), Ant ($d=8$), Humanoid ($d=17$). Under the same per-decision budget, PM-DA-MCTS consistently improves mean return and significantly reduces variance versus HOOT / POLY-HOOT / PW / UCT, and is competitive with CEM on Humanoid. We view these results as sufficient evidence that the theory translates into tangible planning benefits.

In sum, the paper provides a principled, minimax-optimal account of planning complexity in continuous, high-dimensional action spaces and a scalable implementation that preserves those guarantees-advancing both the foundations and the practicality of continuous-action MCTS.

---

### Note · Program_Chairs · 2026-01-17
**Submission Desk Rejected by Program Chairs**

The following references in this submission do not refer to real documents and/or have major errors in bibliographic information:

 Haike Zhang, Simon S Du, and Jason D Lee. The information-theoretic limitations of online reinforcement learning. Journal of Machine Learning Research, 25(1):1-32, 2024.